# Croaking for haste: How long does it take to describe a frog species since its discovery?

**Albert Carné**[1,2,3]*, **Alberto Sánchez-Vialas**[2], **Claudia Lansac**[2,3], **Miriam Moreno-Orosa**[4,5], **Ignacio De la Riva**[2]*

**1** Science and Business S.L., Edificio CITEXVI, Vigo, Spain, **2** Department of Biodiversity and Evolutionary Biology, Museo Nacional de Ciencias Naturales (MNCN), CSIC, Madrid, Spain, **3** Department of Biodiversity Ecology and Evolution, Faculty of Biological Sciences, Complutense University of Madrid, Madrid, Spain, **4** Biodiversity Research Institute (IMIB), University of Oviedo-Principality of Asturias-CSIC, Mieres, Spain, **5** Department of Biology of Organisms and Systems, University of Oviedo, Oviedo, Spain

* albert.carne@mncn.csic.es (AC); iriva@mncn.csic.es (IDLR)

## Abstract

Global biodiversity faces severe anthropogenic threats, with alarming extinction rates projected for the near future. Most of Earth's diversity remains undescribed, meaning countless species are doomed to extinction before being documented. Since current conservation laws typically consider only described species, the lag time in achieving a representative global biodiversity inventory is a crucial issue impeding effective conservation. Amphibians, the most endangered vertebrate class, exemplify the challenge: while the number of threatened species rises, new species descriptions rapidly increase, and hundreds of candidate species are flagged annually. We analyzed all anuran species described from 2000 to 2023 across four biodiversity-rich tropical regions to investigate the time required to describe new frog species. We quantified the time needed to collect the type series, the number and timing of expeditions, the lag between collection and publication, and the total time. Additionally, we explored temporal trends and the effect of selected abiotic variables. The time-lag from the collection of the first specimen to the formal publication of a frog species description ranged from 0.4 to 125.7 years (median = 7.3, mean = 11.3), type series collection from 0 to 104.9 (median = 0.1, mean = 4.5), and description and publication from 0.2 to 54.6 (median = 4.4, mean = 6.8). Alarmingly, the time required to describe new species is globally increasing. Thirty-six percent of species were named within five years of first collection, highlighting the need for continued collecting, biological collections as reservoirs of undescribed diversity, and calling for specimen revision after expeditions. These results raise concerns about the effectiveness of current taxonomic and conservation practices addressing the biodiversity crisis. We call for a global effort to prioritize taxonomic research and discuss taxonomic and conservation approaches. Under current practices, and given the observed timelines, we will lose the race against extinction for many species.

**Data availability statement:** All relevant data are within the manuscript and its Supporting Information files.

**Funding:** AC was funded by MCIN/AEI/10.13039/501100011033, contract for industrial doctorates aid DIN2021-011964. The funders had no role in study design, data collection and analysis, decision to publish, or preparation of the manuscript.

**Competing interests:** The authors have declared that no competing interests exist.

## Introduction

Global biodiversity is being assaulted by Global Change. Wildlife populations are being reduced, contracted, and extirpated; species are becoming extinct at an unprecedented rate, and entire ecosystems are disappearing before we have the opportunity to properly document them [1–4]. The loss of a species is irreversible and its consequences are unpredictable [5]. The highest rates of decline and extinction are occurring in the tropics, hyper diverse areas where scientific knowledge remains limited, but which are known to contain high levels of undiscovered and undescribed diversity [3,6–10].

Species are the fundamental unit to measure biodiversity. Only by recognizing the actual species richness of a given region can we reliably understand large-scale ecological and evolutionary trends (e.g., [11]). How many species are on Earth remain an unresolved question —the Linnean shortfall [12–13], partly because of incomplete sampling and the lack of robust biodiversity characterization and extrapolation approaches [14–15]. However, there are three aspects of consensus: first, the largest proportion of biodiversity is comprised by undescribed species [15]; second, the urgency to discover and describe this diversity has never been greater —naming species is a first step towards their conservation, and overcoming the Linnean shortfall is crucial for understanding distributional patterns and, consequently, for reaching effective conservation planning [16–17]; third, we must explore new taxonomic approaches, as the estimated time required to document the remaining biodiversity far exceeds what may be available due to ongoing biodiversity threats [18–19].

Traditional taxonomic approaches, largely based on morphology, face significant challenges in addressing the urgency of the current biodiversity crisis. Integrative taxonomy, by integrating multiple lines of evidence, emerged to accommodate and unify new concepts and methods from various disciplines to ensure the proper description of taxa, maintain taxonomic stability over time, and prevent taxonomic inflation [18,20]. A "fast-track" taxonomy proposal arose from the pressing need to characterize and describe as much diversity as possible before it disappears, significantly shortening the description time by deliberately omitting previously collected material, focusing on essential diagnostic traits, and reducing the most time-consuming parts of the process (i.e., high-quality images vs. detailed and extensive descriptions, reducing descriptions vs. enhancing diagnoses) [21–22]. However, considering the overwhelming number of candidate species awaiting description worldwide (e.g., [10,23–32], but see [33]), the global richness estimates [15], and the escalating biodiversity crisis, this strategy is likely unable to keep pace with current and projected extinction rates [34], nor to achieve a representative inventory of global species within a reasonable time frame [22]. Some authors, not without receiving criticisms, have attempted the so-called "turbo-taxonomy", which consists in describing large numbers of taxa providing minimal information on each species, sometimes even just a barcoding sequence as the principal basis of species delimitation and diagnosis (e.g., [35]). While this approach may help overcoming the Linnean shortfall, questions arise about the usefulness and reliability of such species "descriptions" [36]. Reliance on mitochondrial DNA or barcodes alone does not provide sufficient evidence for

robust species delimitation, and it also precludes comparison with already described but unsequenced taxa, increasing the risk of redundant nomenclature. Although "fast-track" and "turbo" taxonomy are sometimes seen as synonyms [36], we distinguish them as two different approaches. We consider as "fast-track" those descriptions that, while concise, include all necessary sections to allow a third party to identify the species without the need for a molecular lab (e.g., [37]), while "turbo" are those minimalist descriptions that do not (e.g., [35]).

Regardless of the chosen approach, describing a species involves five steps —the five D's of taxonomy [38]: discovery, delimitation, diagnosis, description, and specimen determination (although not necessarily by the same person). While these steps are conceptually linear, in practice, the process is often iterative, with researchers revisiting and redefining previous steps as new information emerges or as the species boundaries are reassessed. None of these steps can be achieved without previous field work. Fieldwork is time-consuming and, particularly in tropical regions, can be challenging, expensive, and logistically complex. To decide whether a lineage merits to be named, data from different lines of evidence (e.g., morphology, genetics, bioacoustics, ecology) should ideally be gathered, which requires time and cannot always be achieved within a single field trip. Compiling such evidence, preferably including multiple specimens for the type series, can be challenging when cryptic, secretive, seasonal, narrowly-distributed species, or those with low population densities are involved. This is especially true for frogs, where obtaining call recordings —often a source of key diagnostic characters— can be daunting. Specimens representing populations that clearly merit recognition as distinct species can remain unstudied on shelves for years [39], which negatively impacts their conservation prospects, as undescribed species are more vulnerable to extinction [40].

Amphibians are the most endangered vertebrate class worldwide. More than 41% of described species are under threat, and the number of species at high risk of extinction continues to increase [41]. Threats are ubiquitous throughout the amphibian tree of life, with habitat loss and degradation (primarily due to deforestation), climate change, and emerging diseases being the greatest risks to amphibian persistence [41]. There is a geographical bias, with those regions with greater diversity and lesser available information, i.e., the tropics, being the most affected [3,42,43]. Paradoxically, current rates of amphibian species description are high (ca. 150 sp. nov./year) and, during the last decade, hundreds of further candidate species have been flagged globally [10,30–32]. This indicates that we are still far from overcoming the Linnean amphibian shortfall, that the number of threatened species has likely been underestimated, and that many species are facing silent extinction before they are named or even discovered [44].

Here we analyze the type series metadata of all anuran species described since the year 2000 from four tropical regions, to globally and regionally estimate: (1) how long it takes to describe a frog species since the first type specimen is collected; (2) how long it takes to get the complete type series for describing a frog species; (3) how long it takes to describe a frog species since all the type material has been collected; (4) how many field trips are needed to describe a frog species; (5) whether description times are decreasing or increasing over time; and (6) the abiotic factors that are related to some frogs being described in a notably shorter time than others, and (7) how these factors have evolved over time.

## Materials and methods

This study did not involve any experiments or direct interaction with live animals. All data analyzed were obtained exclusively from previously published literature. Therefore, ethical approval from an Institutional Animal Care and Use Committee (IACUC) or other ethics board was not required for this research.

### Region selection, data acquisition and curation

We selected four regions across the globe based on two criteria: high frog species richness (i.e., > 300 described species), and an exponential rate of accumulated frog species descriptions. The regions chosen were Ecuador, India, Madagascar, and Melanesia. These regions are taxonomically independent and, in principle —and largely in practice— do not share

 

taxonomists, which helps control for regional biases in research practices. By selecting geographically scattered regions, we aim to investigate the time span of the species description process on a global scale, diluting regional biases to obtain a more accurate timeframe that can be considered globally applicable.

We retrieved the list of described frog species for each region, up to December 2023, from Amphibian Species of the World (ASW; [45]), filtering by higher taxa (Anura) and country, and retaining only native species (i.e., discarding those introduced). Since Melanesia includes two countries —Indonesia and Papua New Guinea—, and parts of Indonesia extend well beyond the region, we could not easily obtain the species list from the online database. Therefore, in accordance with the definition of the Melanesian region provided by Oliver et al. [31], we compiled the species list by selecting "Indonesia-Papua Region", "Papua New Guinea", "Solomon Islands", and "Fiji" (which also includes smaller nearby islands) in ASW.

To determine the species description lags for a contemporary timeframe, we filtered each regional species-list by year, retaining only those species described from the year 2000 (included) onward. We decided to exclude data predating 2000 as software and genetic data —assumed to accelerate species discovery, diagnosis, and description— became widely available and utilized (with regional variability) only in the 21st century. Genetic techniques have become crucial for describing species, but also for detecting cryptic species and screening regions (and public collections) to unveil overlooked and cryptic diversity (e.g., [10,30]).

For each species, we retrieved the publication and all collection dates of the type series (holotype and paratypes) from the original descriptions, excluding additional and referred specimens. We only considered unique collection dates; that is, if multiple specimens were collected on the same day, we registered only one date. To deal with imprecise dates (e.g., 5–7 December 2008, February 2007) we used the midpoint date (e.g., 6 December 2008 and 14 February 2007, in the provided examples).

## Lag times

We made several necessary assumptions regarding the species description process, and defined three time periods: "overall description process", "collection", and "description *sensu stricto*". We acknowledge that defining these periods and making these assumptions is an oversimplification, but it is necessary to draw conclusions (see Discussion).

We consider the "overall description process" as the time span between the collection date of the first (i.e., the oldest) specimen included in the type series and the publication date of the article introducing the new species to the scientific community. Within this overall process, we differentiate the two other sub-periods, the "collection" and the "description *sensu stricto*".

The collection period refers to the fieldwork conducted to gather the entire type series, extending from the collection of the first specimen to the last. We assume that all type specimens were considered essential for the species description, which was delayed until the completion of the type series, and that the inclusion of both recently collected and older specimens was not arbitrary. Additionally, we regard the first collected type specimen as the first scientific encounter with the species (i.e., the moment a researcher truly discovers the species), even if it was not immediately recognized as a new taxon in the field and this insight occurred years later.

The description *sensu stricto* is the process encompassing taxon delimitation, diagnosis, description, and determination (*sensu* [38]), spanning from the collection of the last specimen of the type series to the publication date. At this point, we assumed that once the last specimen of the type series was collected, the description *sensu stricto* started.

To summarize and visualize the three periods, we calculated their durations in years for easier interpretation and representation. Using the *base* package in R [46], we obtained central tendency statistics including the mean, trimmed mean (mean after excluding 10% of extreme values from both ends to reduce the influence of outliers), and median, along with position statistics such as the minimum and maximum. Using the *psych* package [47], we then calculated measures of asymmetry, including skewness (an asymmetry coefficient where positive values indicate a long right tail, meaning most observations are concentrated below the mean with some high values) and kurtosis (a measure of tail weight and

peakedness where high values indicate heavy tails and a sharp central peak compared to a normal distribution). Using the *psych* package, we also calculated dispersion measures such as the Standard Deviation (S.D.), Median Absolute Deviation (M.A.D.; a robust dispersion measure; calculated as the median of absolute differences from the median, less sensitive to outliers than S.D.), range, and Standard Error (S.E.). We used the *ggplot2* package [48] to represent the results using violin and boxplots. This process was conducted independently for each genus and region, followed by merging the four regional datasets to obtain global estimates.

### Number of field trips and temporal patterns of type specimen collection

To estimate the number of field trips required to collect the entire type series for each species description, we calculated the time span between the sorted collection dates of the specimens in the type series. We considered as distinct field trips those collection events separated by more than one month (i.e., 30 days).

To estimate global and regional temporal patterns in the collection of type specimens (i.e., the number of field trips conducted to collect type specimens each month and year), we calculated the number of specimens collected per month and year, and represented the collection frequencies using heatmaps with the *ggplot2* package.

### Effect of abiotic variables on the description lags

To understand why certain species were described faster than others, we investigated the effect of several abiotic variables on the description *sensu stricto*. We did not analyze the effect of these variables on the overall and collection processes, as the values of these variables are typically determined post-collection.

We included three categorical variables: (1) whether genetics were involved in the species description (YES/NO), (2) the region, and (3) the genus; also, we considered five continuous variables: (1) the number of authors of the description paper, (2) the number of specimens in the type series, (3) the number of species described in each paper, (4) the number of field trips required to collect the type series (see above), and (5) the number of species currently recognized in the genus.

Before fitting the models, we ensured that no numerical variable had a correlation greater than 70% with any other to avoid multicollinearity. For each of the four regions and the global analysis (i.e., combining the four regions into a single dataset), we fitted generalized linear mixed models (GLMM) using the *glmmTMB* R package [49]. In the global model, we included "genus" and "region" as random effects to account for the hierarchical structure of the data. We excluded "region" in the regional models. All models used a Gaussian family with an "identity" link function. We log-transformed the response variable to meet or improve the model assumptions.

To further assess the significance of the relationships, we performed an analysis of variance using the *Anova()* function from the *car* R package [50], which allowed us to evaluate the influence of the predictor variables without considering the factor levels.

We evaluated the model performance using the *DHARMa* package [51], by obtaining the simulated residuals from the model to assess the deviation from the expected distribution, uniformity, outliers, and dispersion. Additionally, we fitted null models (i.e., only including the random effects) and compared these with the empirical models to determine whether the inclusion of fixed effects in the empirical model, provided a significantly better fit to the data. We conducted this comparison using *anova()* from the *stats* package. Finally, we used the *effects* package [50] to represent the effects of the variables.

### Temporal evolution of variables

For each region and globally, we evaluated the temporal evolution of several variables including the overall description process, the collection, the description *sensu stricto*, the number of authors, the number of species described each year, the number of taxonomic papers published annually, the number of species described in each paper, the number of type specimens in each species' type series, the percentage of species described using genetics each year, and the number of field trips undertaken to collect the type series.

To analyze the temporal evolution of these variables, we averaged the yearly values for each region and globally (e.g., the mean time invested in the collection of type specimens for each species described in 2006). Additionally, we counted the number of species described each year and the number of those that used genetic data, in order to compute the percentage of species descriptions incorporating genetic evidence per year.

We used the *ggplot2* package to generate all plots, applying local regressions (LOESS) to fit curves to the data for visualization purposes. To assess the temporal significance of the aforementioned variables, we calculated *p-values* using linear (LM, lm()) or generalized additive models (GAMs, gam()) from the *stats* package with default parameters [46]. We selected the best method based on the comparison of their coefficient of determination ($R^2$).

## Results

### Species described since the year 2000 and lag times

Nine hundred and twenty-four frog species have been described across the four analyzed regions between 2000 and 2023. Melanesia leads with 286 newly described frog species, followed by Madagascar with 215, Ecuador with 213, and India with 210. This represents an average description rate of ca. 10 new species per region each year. After excluding unavailable papers and species with unreported collection or publication dates, we analyzed a total of 2,981 unique collection dates from 896 species; 269 species from Melanesia, 215 from Madagascar, 208 from Ecuador, and 204 from India (S1 Table and S2 Table).

Thirty-six percent of the analyzed species were described within five years of the collection of the first type specimen, and ca. 26% of the species between five and ten years of the first collection. The remaining ca. 40% of the species were described between 10 and 125 years after the first type specimen collection (Fig 1).

The mean and confidence interval for the global overall description process (i.e., from the collection of the first type-specimen to the paper publication date) is 11.3±0.8 years, with 4.5±0.7 years dedicated to collecting the type series and 6.8±0.5 years focused on the description *sensu stricto*. However, the distribution is heavily right-skewed, indicating that while some species experienced long delays, most species were described in a shorter time. This is reflected in the medians, which are substantially lower: 7.3 years overall, 0.1 years for collection, and 4.4 years for description and publication (see Fig 2 and Table 1 for central tendency, position, asymmetry, and dispersion statistics).

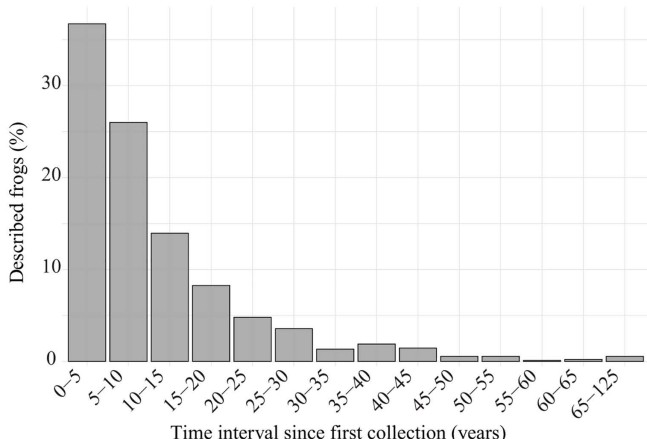

**Fig 1. Percentage of frog species described since their first collection.** Percentage of frog species described as a function of the time elapsed since the collection of the first specimen.

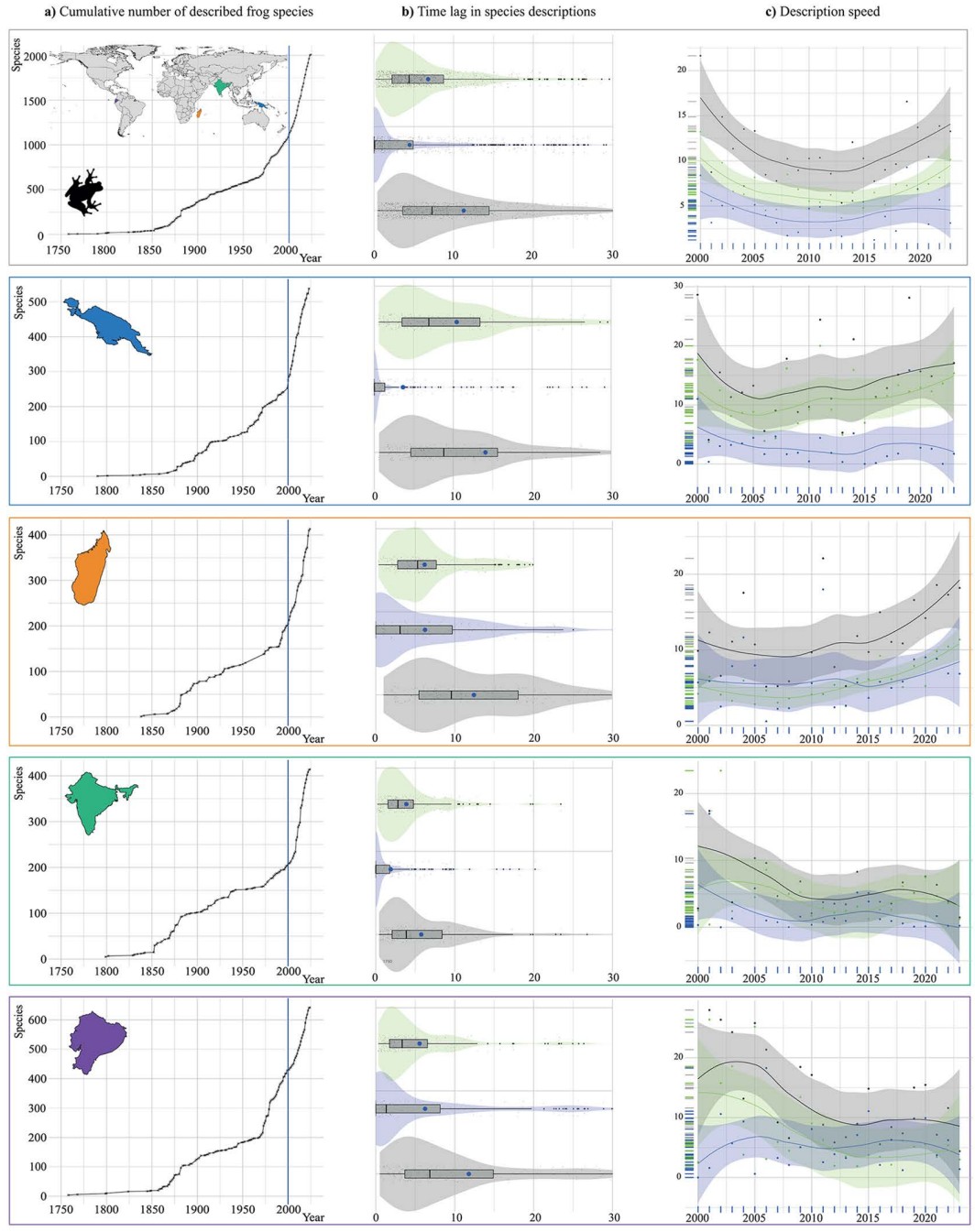

**Fig 2. Time lags and trends in anuran amphibian species descriptions. (a)** Cumulative number of frog species described over time. **(b)** Time lags in the description process. **(c)** Average description speed per year. Green: description *sensu stricto*, blue: collection, gray: overall process. The world map showing country boundaries was obtained from OpenDataSoft (https://public.opendatasoft.com/explore/dataset/world-administrative-boundaries/information/) and is licensed under the Open Government Licence v3.0, which is compatible with the Creative Commons Attribution License (CC BY 4.0).

**Table 1. Summary statistics (in years) for each analyzed region across the different subperiods of the species description process.** The table presents the minimum (Min.), mean, standard deviation (S.D.), 10% trimmed mean (Trim. mean), median, median absolute deviation (M.A.D), maximum (Max.), range, skewness (Skew), kurtosis, standard error (S.E.), and 95% confidence interval for the mean (Interval).

| | GLOBAL | Ecuador | India | Madagascar | Melanesia |
|---|---|---|---|---|---|
| **Species** | 896 | 208 | 204 | 215 | 269 |
| **Overall description process** | | | | | |
| Min. | 0.42 | 0.5 | 0.42 | 0.99 | 0.61 |
| Mean | 11.26 | 11.79 | 5.77 | 12.43 | 14.08 |
| S.D. | 12.63 | 12.58 | 6.16 | 10.48 | 16.18 |
| Trim. mean | 8.91 | 9.43 | 4.78 | 11.25 | 11.01 |
| Median | 7.28 | 6.85 | 3.87 | 9.57 | 8.81 |
| M.A.D | 6.59 | 5.86 | 3.4 | 7.84 | 7.59 |
| Max | 125.74 | 82.95 | 60.04 | 110.86 | 125.74 |
| Range | 125.32 | 82.45 | 59.62 | 109.88 | 125.13 |
| Skew | 3.5 | 2.03 | 4.08 | 4.16 | 3.22 |
| Kurtosis | 20.48 | 5.17 | 28.75 | 34.34 | 15.31 |
| S.E. | 0.42 | 0.87 | 0.43 | 0.71 | 0.99 |
| Interval (95%) | 11.26±0.82 | 11.79±1.71 | 5.77±0.85 | 12.43±1.40 | 14.08±1.93 |
| **Description *sensu stricto*** | | | | | |
| Min | 0.18 | 0.33 | 0.18 | 0.35 | 0.57 |
| Mean | 6.79 | 5.54 | 3.88 | 6.19 | 10.45 |
| S.D. | 7.52 | 6.8 | 3.6 | 4.46 | 10.3 |
| Trim. Mean | 5.33 | 4.15 | 3.27 | 5.56 | 8.43 |
| Median | 4.41 | 3.35 | 2.81 | 5.3 | 6.92 |
| M.A.D | 3.74 | 2.83 | 2.37 | 3.72 | 6.33 |
| Max | 54.62 | 51.48 | 23.44 | 19.91 | 54.62 |
| Range | 54.45 | 51.14 | 23.26 | 19.56 | 54.05 |
| Skew | 2.85 | 3.27 | 2.21 | 1.21 | 1.96 |
| Kurtosis | 10.57 | 14.09 | 6.8 | 0.82 | 3.93 |
| S.E. | 0.25 | 0.47 | 0.25 | 0.3 | 0.63 |
| Interval (95%) | 6.79±0.49 | 5.54±0.92 | 3.88±0.49 | 6.19±0.60 | 10.45±1.23 |
| **Collection** | | | | | |
| Min | 0 | 0 | 0 | 0 | 0 |
| Mean | 4.47 | 6.25 | 1.89 | 6.24 | 3.64 |
| S.D. | 9.93 | 10.58 | 5.13 | 9.96 | 11.54 |
| Trim. Mean | 2.2 | 3.92 | 0.75 | 4.52 | 0.92 |
| Median | 0.08 | 1.32 | 0 | 3.07 | 0.01 |
| M.A.D | 0.12 | 1.96 | 0 | 4.56 | 0.02 |
| Max | 104.93 | 79.97 | 55.84 | 104.67 | 104.93 |
| Range | 104.93 | 79.97 | 55.84 | 104.67 | 104.93 |
| Skew | 4.94 | 2.84 | 6.54 | 5.05 | 5.51 |
| Kurtosis | 35.46 | 11.73 | 59.3 | 43.02 | 36.14 |
| S.E. | 0.33 | 0.73 | 0.36 | 0.68 | 0.7 |
| Interval (95%) | 4.47±0.65 | 6.25±1.44 | 1.89±0.70 | 6.24±1.33 | 3.64±1.38 |

We found significant disparities among regions. India leads in the speed of species descriptions, averaging 5.8±0.9 years of overall process, 1.9±0.7 years for collection, and 3.9±0.5 years for description. Ecuador follows with an overall process time of 11.8±1.7 years, 6.3±1.4 years for collection, and 5.5±0.9 years for description. Madagascar closely trails, with an overall process of 12.4±1.4 years, comprising 6.2±1.3 years for collection and 6.2±0.6 years for description. Finally, the region that takes the longest is Melanesia, with an overall process time of 14.1±1.9 years, including 3.6±1.4 years for collection and 10.5±1.2 years for description (Fig 2 and Table 1).

In all three periods (i.e., overall, collection, and description) and across all five regions (i.e., four independent regions, plus global), we obtained high standard deviation values, with values exceeding the mean value in 11 out of 15 cases. This indicates a significant dispersion of the data around the mean. The median values consistently fell below the mean, suggesting a skewed distribution. Skewness values were greater than zero in all cases, pointing to positive asymmetry. Kurtosis values consistently exceeded zero, signaling heavy tails and the presence of more extreme values (outliers) that would typically occur in a normal distribution. Additionally, the standard error values remained low in all cases, especially in comparison to the standard deviation, indicating that the obtained mean is a fairly reliable estimate of the population mean.

## Number of field trips and temporal patterns of type specimen collection

Half of the species (476; 53.1%) described between 2000 and 2023 required a single field trip to collect the entire type series, with an average of eight collected specimens per species. Two trips were required for 20.3% of species, three for 11.7%, and four for 6.1%. The remaining ca. 8% were described on the basis of collections made over between five and 23 field trips.

Considering the four regions, species descriptions between 2000 and 2023 were based on type specimens collected between 1893 and 2023, with the majority collected between 1999 and 2016, with some regional differences (S1-S5 Fig).

In Melanesia and Ecuador (S4-S5 Fig), type specimen collection was relatively evenly distributed throughout the year (defined as each month having at least 40% of the number of specimens collected in the peak month). In contrast, Madagascar and India exhibit marked seasonal peaks in collection activity, with certain months showing higher numbers of type specimens collected than others (S2-S3 Fig).

## Effect of abiotic variables on the description lags

Empiric models differed significantly from the null models and presented lower AIC values, indicating that the models are better than randomness and that the fixed variables have a genuine effect on the data (S3 Table). All models met the model assumptions, except for Ecuador, where minor violations were observed (S6 Fig). Adding random effects ("genus" in all cases, and both "genus" and "region" in the global analysis) significantly increased the percentage of variance explained by the models (S4 Table).

In all regions except Ecuador, the number of authors had a significant effect on species description time, with an increase in the number of authors associated with a faster description in the global model, Ecuador, and India, and with slower description times in Madagascar and Melanesia. The number of type specimens did not have a significant effect in any region. The number of described species within a genus was marginally significant in Madagascar, where richer genera are associated with faster species descriptions. This variable was not significant in other regions. The number of field trips significantly influenced the description times in the global model and in India, was marginally significant in Ecuador and Madagascar, and showed no significant effect in Melanesia. Across all regions, however, an increase in the number of field trips to collect the type series was associated with shorter description times, perhaps because much of the work had already been done and only complementary data, such as vocalizations, were needed. The number of species described per paper was significant in the global model, Ecuador, and Melanesia, but not in Madagascar or

India. An increase in the number of species per paper was associated with longer description times in all regions except in Madagascar, where the effect was the opposite (but not significant). Incorporating molecular data was associated with longer description times in all regions except Madagascar, but this effect was significant only in the global model and in India (S7 Fig, S4 Table).

## Temporal evolution of variables

Globally, the overall description process decreased from 16 years in 2000 to 7.2 years by 2012. However, after 2012, the description lag increased again, reaching a current value of ca. 11 years (Fig 3). Both collection and description *sensu stricto* followed similar declining trends until 2012, after which the description time rose significantly, while collection time remained stable. Statistical analyses showed that both overall description time and description *sensu stricto* varied significantly with the year (p-value$_{GAM}$: 0.023 and 0.015, respectively), while collection time remained stable (non-significant).

Ecuador, India, and Melanesia broadly follow the global pattern in the overall description (p-value$_{GAM}$: 0.01, 0.03, and n.s., respectively), in the description *sensu sticto* (p-value$_{GAM}$: 0.01, n.s., and n.s. respectively), and in the collection processes (n.s.). In contrast, Madagascar shows a considerable increase in both the overall description process and description *sensu stricto* (p-value$_{GAM}$: 0.03 and <0.001, respectively), as well as a moderate, non-significant increase in the collection time (S8-S11 Fig).

The number of authors contributing to species descriptions has significantly increased globally (p-value$_{LM}$: <0.001). In Ecuador and India, the number rose from ca. two to five authors, where it appears to have plateaued (p-values$_{GAM}$: <0.001), while in Madagascar and Melanesia, it continues to rise (p-values$_{GAM}$: <0.006) (S8-S11 Fig).

The number of taxonomic papers published globally each year increased from ca. 12–23, plateaued until 2020, and then declined (p-value$_{GAM}$: 0.01). Ecuador, India, and Melanesia follow similar patterns (p-values$_{GAM}$: <0.001, <0.001, and n.s.). In contrast, Madagascar shows high variability, with an overall mean of ca. five annual papers, with no significant trend (S8-S11 Fig).

The proportion of taxonomic papers incorporating genetic data has significantly increased globally, reaching 90% by 2022 (p-value$_{GAM}$: <0.001). Madagascar shows the earliest widespread usage, with 100% of papers incorporating genetic data since 2004, with few exceptions (p-value$_{GAM}$: <0.001). In India, the use of genetics remained low (ca. 25%) until 2010, followed by an increase to the current 100% (p-value$_{GAM}$: <0.001). In Ecuador, usage has increased steadily, currently exceeding 80% (p-value$_{GAM}$: <0.001). In contrast, Melanesia lags behind, with only ca. 40% of papers incorporating genetic data (p-value: 0.005) (S8-S11 Fig).

The number of species described annually increased globally from ca. 25–45 until 2010, when it plateaued (p-value$_{GAM}$: 0.01). Ecuador shows a similar trend, reaching a plateau of ca. 12 species per year by 2018 (p-value$_{GAM}$: <0.001). India shows a non-significant steady increase until 2012, followed by a decline. Madagascar and Melanesia show great variability with an overall non-significant continuous increase without reaching a plateau in Madagascar and an overall decrease in Melanesia (S8-S11 Fig).

The average number of species described per paper shows a globally non-significant positive trend. Ecuador and Madagascar show similar patterns, while in India and Melanesia, the number increased until ca. 2013, followed by a decline (S8-S11 Fig).

The number of type specimens in a type series has globally decreased from ca. 20 to fewer than 10 (p-value$_{LM}$ 0.005). A similar pattern is observed in Melanesia (p-value$_{LM}$ 0.003). Ecuador and India show no significant trends. In Madagascar, the overall trend is a steady decrease, although some recently described species have reversed that trend (p-value$_{GAM}$ 0.004) (S8-S11 Fig).

The number of field trips required to collect the entire type series globally shows huge dispersion, with an overall stable trend between 1.5 and 3 field trips required to collect the type series (S8-S11 Fig).

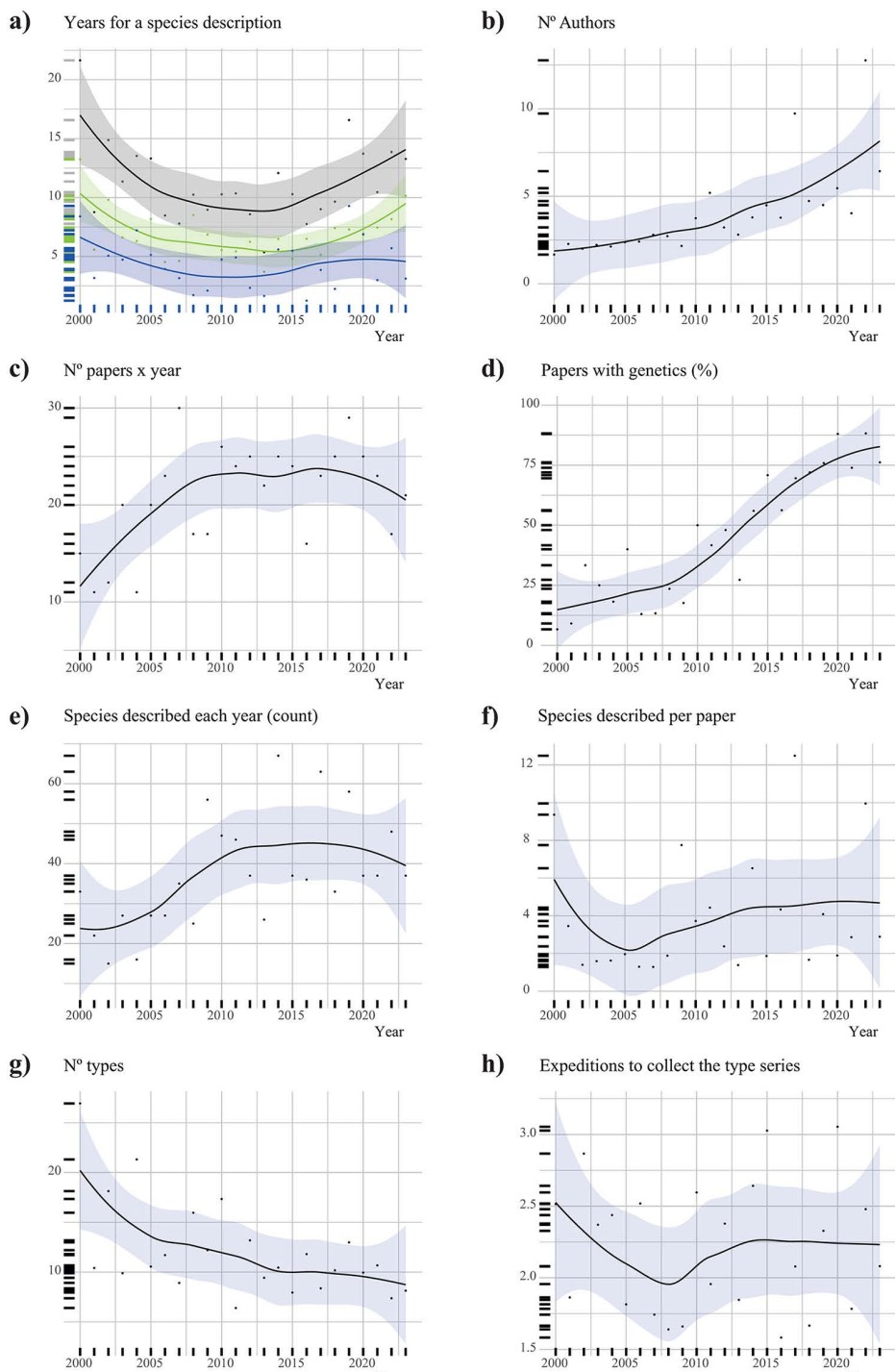

**Fig 3. Temporal evolution of variables in the four analyzed regions.** (a) Overall description process (black), collection (blue), and description *sensu stricto* (green), (b) number of authors in each species description (mean), (c) number of published taxonomic papers (count), (d) Taxonomic papers that include genetics (percentage), (e) number of species described each year (count), (f) number of species described in each paper (mean), (g) number of type specimens in the type series (mean), (h) number of field trips conducted to collect the entire type series (mean).

## Discussion

### Anuran species descriptions amid a biodiversity crisis

Taxonomists have named more than 2.3 million species [52] at a pace of 20,000 species descriptions per year [53]. However, this represents only a small fraction of the world's predicted species richness [15]. In the past century, we have come to realize that our planet is facing a biodiversity crisis, while the majority of species remain undiscovered and undescribed. Unless we reverse the current trend of threats, we may be among the last generations with the opportunity to explore and document Earth's biodiversity [39,54].

Here, we evaluated the time required to describe a frog species with a focus on four megadiverse tropical regions. Despite being the most endangered class of vertebrates [41], amphibians are being described at remarkably high rates, and the numerous candidate species flagged worldwide suggest that true amphibian richness —and consequently the number of threatened species— has been, and remains, underestimated [10,30,32,55].

Our results suggest that the mean description time from the collection of the first type specimen to publication is 11.3 years per species, with 4.5 years dedicated to the type series collection and 6.8 years to the description *sensu stricto*. We found similar values across three of the four analyzed regions, with India showing lower values (see Results). Fontaine et al. [39] found similar "shelf" figures for tetrapods analyzing a random set of species described in 2007. Our results indicate that, for frogs, the description times are not decreasing at the global level. While continental regions (e.g., India and Ecuador) exhibit a trend of declining description times, insular regions (e.g., Madagascar and Melanesia) show an increase. This pattern may be explained by the increasing presence and activity of local taxonomists in continental countries, while in the studied insular regions, taxonomy has historically been led by foreign institutions, which are experiencing a reduction in the number of specialists [56]. Overall, our estimate of the global pattern is U-shaped, with current description times being longer than those in 2012 but shorter than those in 2000.

Thirty-six percent of frog species described since 2000 were named within five years of first collection, indicating that a substantial proportion of recently described species rely on newly collected material. This highlights the magnitude of the Linnean anuran shortfall and the importance of maintaining active fieldwork and collecting —particularly in poorly known regions— at a time when logistical, political, and regulatory barriers to collecting and vouchering are increasing. In contrast, the remaining 64% of species were collected earlier —up to 125 years ago— and described much later. This suggests that new species are not always recognized as such in the field, or, when they are, their formal description may be delayed by the limited resources and capacity of (the few) taxonomists to process the sheer volume of undescribed species, or because other research priorities are often more academically rewarded. It also reflects that targeted sampling aimed at collecting specific, suspected new species is relatively uncommon. These findings also support the critical role of public scientific collections as reservoirs of undocumented diversity, while also raising concerns about the limited personnel, resources, and time available to examine the collected material. Public collections contain a wealth of overlooked specimens (e.g., [57–61]). In fact, as Bebber et al. [62] noticed, a significant percentage of the world's undescribed species have already been collected and lodged in museums, but await further scientific investigation and description. In the context of the current biodiversity crisis, some authors have suggested prioritizing the collection and preservation of as many specimens as possible as testimonies of a vanishing biodiversity, ensuring future generations can describe species that may already be extinct [39–63]. Collecting efforts are essential for addressing the geographical and taxonomic gaps that still exist. To maximize their contribution to biodiversity discovery, collecting should be paired with proper specimen processing (including tissue sampling for genetic analysis) and thorough taxonomic revision to promptly detect candidate species and prioritize their description. To describe as many species as possible while they still exist, it is essential to reverse decades of funding cuts to taxonomy. Increased investment would support the creation and maintenance of much-needed taxonomic positions, expand sampling efforts, and enable thorough scrutiny of both recent and historical collections.

Tropical regions harbor most of Earth's terrestrial biodiversity, yet they remain highly threatened and among the least scientifically understood [64–65]. Large-scale molecular screenings are still relatively rare, but those conducted have

revealed high numbers (even hundreds) of potential new species—enabled by comparisons with previously sequenced taxa [10,30,32,55]. Despite growing use of molecular data in species descriptions over the last decades, it has yet to become a standard practice (see results; [53]). Our results suggest that incorporating molecular data into frog descriptions significantly increases description time in some regions. While adding any procedure to the species descriptions inherently increases the time required, the increase of time can also be attributed to cryptic species, which may remain undescribed on shelves for years until identified molecularly. Sequencing type specimens during species descriptions offers benefits for taxonomy and biodiversity science. DNA-barcodes linked to identified and re-examinable type specimens establish a reliable connection between genetic data and physical vouchers from the outset [66]. This enables unambiguous specimen assignment to described species often without taxonomic expertise—except in cases of introgression, hybridization, or human error [67]. Therefore, by leveraging new methodologies and increasingly affordable molecular tools, the widespread use of DNA barcodes to screen collected material should ideally become a standard practice among taxonomists where feasible, especially in poorly surveyed regions [68]. Genetically characterizing the collected material and publicly releasing the georeferenced sequences would not only highlight candidate species — thereby promoting their formal description— but also provide valuable data for other disciplines such as biogeography, evolutionary biology, and ecology. Such efforts could help reduce the Wallacean shortfall [12] by revealing new localities (e.g., metabarcoding) and improving our understanding of intraspecific diversity (e.g., macrogenetics; [69]). Although the ideal species description involves partial sequencing to allow DNA-based identification, genetic comparison, and phylogenetic contextualization, the use of molecular data is not strictly necessary. We acknowledge that socio-economic constraints and legal restrictions on specimen or tissue export in many countries unfortunately limit the widespread implementation of these molecular methods.

Given the number of species still awaiting discovery and description, it is evident that we need more effective solutions for inventorying and protecting Earth's amphibian fauna before many species vanish. Over the average 11-year span than takes to formally describe a frog species since the first specimen is collected, vast areas of currently suitable habitats are likely to disappear if current rates of habitat destruction continue [70–71].

There is an undeniable urgency to describe as many species as possible. However, speeding up the process of naming species should not compromise the quality of species hypotheses [22,36]. Adopting streamlined species description methodologies that concisely focus on essential diagnostic traits without compromising the rigor of species descriptions (e.g., "fast-track taxonomy" in the sense explained above) and integrating molecular data to support rapid identifications can improve the efficiency of species discovery and description. Although integrative taxonomy has long been practiced by many taxonomists, it is in the last decades that taxonomy has truly become multidisciplinary, integrating approaches and techniques from disciplines such as genetics, biogeography, bioacoustics, morphometry, and osteology (CT scans) to enhance accuracy and provide more information about species. Overcoming the Linnean shortfall while ignoring the Wallacean and other shortfalls (see: [72]) makes little sense if the ultimate goal is to conserve species. But how much complexity is necessary for a species description? Must every shortfall be addressed or every bone be described in the initial process of description? Certain standards are clearly necessary to ensure taxonomic descriptions remain consistent, comparable, and facilitate the work of future taxonomists. Although new methods have emerged to disentangle taxonomic challenges derived from poor descriptions or poorly preserved specimens (e.g., [73]), we still face the consequences of brief historical descriptions, scattered (or lost) type material, and imprecise type localities, which deter anyone from tackling certain taxonomic decisions [21,74].

## Taxonomic data

A foundational task of taxonomy is to compare the variability of target specimens with that of related described species to assess whether the observed differences justify their recognition as a distinct unit [75]. Another challenge in taxonomy is that much of the original biological materials used in species descriptions are often scattered across museum collections

worldwide, making it difficult and expensive to revisit and re-study. Additionally, extensive biodiversity data literature exists, comprising different sets of information. While genetic data are now centralized and easy to retrieve (e.g., GenBank, [76]; BOLD, [77]), other crucial data produced during species descriptions are either absent from accessible repositories, incomplete, or insufficiently standardized, limiting their reuse [53–54].

A previously suggested strategy to enhance data accessibility and thereby accelerate species descriptions, without compromising species hypothesis, is "wikifying" taxonomic data (i.e., cybertaxonomy [21,53,78–80]). The formal description would continue to follow (unless a better solution is found) established taxonomic procedures (i.e., peer-reviewed publications that comply with the International Code of Zoological Nomenclature). However, data about a species or candidate species would be accessible on a centralized digital platform for immediate download and re-use, contributed by ORCID-identified researchers or drawn from referenced literature. If centralized, this system would make it easier to find all information about a taxon, substantially shortening the time needed to compile datasets for species comparisons, and facilitate the continuous updating of species descriptions and data (e.g., DNA sequences, traits, and distributional data) before, during, or after their formal description. This strategy would, on one hand, allow species to be described following a (to be defined) minimum standard to ensure reliable descriptions, with the possibility of later adding information to update them ( [21,53]; and references therein). On the other hand, it would draw attention to undescribed lineages and raise awareness among colleagues working on, or possessing information about, the same taxa [21,81,82]. Consequently, it would not be necessary to delay naming species to provide the most complete description possible. Species and candidate species could be included in the conservation scheme from the outset, and although the act of naming itself remains static, their descriptions and biological content could remain dynamic and be updated over time. In addition, this approach could promote the release of data that, due to their magnitude or nature, are not usually published but are nonetheless valuable (e.g., morphological measurements of already described species). Since a small number of journals publish most taxonomic papers, these leading journals —as suggested by Miller et al. [83]— could coordinate with others to request standardized taxonomic data (e.g., morphometric values, variable labeling, coordinates) and routinely upload it to digital platforms (see a proposed pilot submission template for taxonomic data: [53]). Once a shared minimum standard for reliable descriptions and a unified data submission format are established, other journals and authors could follow the same templates, ensuring consistency across the field. Ultimately, as in many other fields, AI is expected to play an important role in several aspects of taxonomic practice, most likely contributing to the acceleration of species descriptions. However, discussing the potential developments of this emerging tool for taxonomy is beyond the scope of this paper.

Species can be protected through direct conservation efforts by targeting specific species or indirectly by safeguarding their habitats. Undescribed species are not assessed by the IUCN and are generally excluded from biological inventories used to prioritize conservation areas. With the current figures of species descriptions, if we must wait until a species is formally named to protect it, we will lose the race against extinction for many of them. Because of the multiple constraints that can delay formal naming, taxonomists often describe species that have been identified as such for decades. In some cases, given the alarming rate of habitat destruction, it happens that by the time a name is finally given, the type locality has been severely altered or lost (e.g., [84–85]). This highlights a critical gap in conservation: undescribed species constitute the majority of Earth's biodiversity, but lack formal recognition and, consequently, legal protection, making them more vulnerable to extinction [40].

Therefore, a major challenge faced by undescribed species is that current conservation legislation is generally (named) species-centered. Since descriptions are slow and take increasingly longer, many species do not (and will not) have any form of protection unless they happen to occur within a protected area by chance. Using cybertaxonomy could enable including undescribed species in the conservation scheme from the outset and/or enhance a region-centered conservation approach (i.e., indirect conservation). In the latter, we propose accounting for species that have been known for

years but remain unnamed—many of which we have substantial data on (e.g., [37]). These species could be used to prioritize conservation areas based on species "richness", aligning with global targets such as the 30 × 30 initiative, which aims to protect 30% of the planet's land and oceans by 2030 [86]. This approach would prioritize long-term conservation efforts by considering a more realistic diversity inventory. To achieve this, we suggest publishing all available information on undescribed species, such as DNA barcodes and distribution data, to make it available for regional or global studies. We acknowledge that this approach, in some occasions, could inflate the species inventory of a given area. However, the worst outcome would be conserving divergent conspecific lineages, which, in any case, represent important genetic variability that should be preserved.

## Taxonomy crisis

Taxonomy is a foundational science. Almost all biological studies take place downstream ( [38,54]). Contradictorily, is an undervalued biological discipline [63]. Taxonomic papers receive few citations and are often published in low-impact journals [87–88] threatening the careers of early-career researchers with low citation indices [89–90]. Additionally, funding for taxonomy-focused projects is steadily declining [54,78,91,92]. To secure funding and publication in high-impact journals, these projects often need to be framed within cutting-edge methodologies or broader questions. This can result in using a sledgehammer to crack a nut, ultimately extending description times.

Whether taxonomists are facing extinction is hotly debated [56,63,88,93–95]. Some authors argue that the number of authors involved in species descriptions has increased exponentially [56,96–98]. Others contend that, despite this author inflation, not all authors listed in taxonomic papers can be considered taxonomists [94,98]. On a global scale, there is a regional turnover, with historically leading taxonomic institutions losing taxonomists, while other regions are increasing their numbers [7,56]. Our results confirm that the number of authors involved in frog species descriptions has been steadily increasing. However, we found a globally significant relationship between the increase in the number of authors and a reduction in the description time. These results contradict the presumed undesirable effects of author inflation hypothesis, indicating that task distribution among authors may lead to more efficient species descriptions. However, we did not find this relationship in all regions, suggesting that such improved efficiency may depend on the particularities of each research group.

Our results suggest that, since ca. 2010, the number of published frog taxonomical papers, the number of frog species described each year, and the number of species described per paper have remained stable. This contrasts with the growth in scientific productivity, partially driven by advancements in computational power, new and affordable sequencing technologies, and data analysis software [53,73,99–102]. Unchanged species diagnosis and description procedures have been flagged as the main cause [21,53]. Moreover, the number of species described per taxonomist has generally declined. While some authors have attributed this trend to a lack of new species to describe [21,93,96,98], the large number of candidate species flagged annually and current rates of species descriptions suggest otherwise. This decline may reflect a growing shortage of taxonomic specialists and an increased participation in taxonomy of researchers from other fields [94,98,103]. This trend may be further exacerbated by the fact that describing a new species, while inherently engaging for most biologists, it is little rewarded. As a result, despite not being taxonomists, some researchers venture into describing one or a few species.

Working in consortiums has been proposed as a strategy to increase the number of species described per article, thus improving the impact metrics of both articles and journals. While this strategy may increase impact metrics, an issue remains: who will be the first author? Is it more advantageous for early-career researchers to be the first author of a less-cited article or to be one among many contributors in a consortium article? Regarding description time, our results suggest that while an increase in the number of authors is associated with shorter description times at a global scale, a higher the number of species described per paper (which is expected to increase the impact metric) has the opposite effect, with more species leading to longer times required to describe each one.

## Study caveats

We had to define two subperiods in the species description process and make oversimplified assumptions to analyze data and draw general conclusions. First, the oldest collected specimen in the type series may not represent the species' first scientific encounter, as earlier specimens might have been overlooked (deliberately or not) or stored elsewhere. Even if it was the first encounter, it might not have been recognized as new in the field. In some cases, collectors are not researchers, and the collected material is often not examined immediately after fieldwork. Second, a species can be described using only the holotype (or even part of it). Since not all specimens in the type series are essential for the species description, the process might not be delayed until their collection. Third, including ancient specimens in the type series of species description was not always strictly necessary, and their inclusion extended the calculated lag time between the first collection and publication. Fourth, the description *sensu stricto* rarely begins immediately after the last type specimen is collected. There are multiple factors that can delay the description of a species for decades. Lastly, we acknowledge that our figures may contain a level of error affecting all species in different degrees, which we did not account for: the time span between manuscript submission and publication. This interval varies depending on the journal and the length of the review process, which can sometimes be lengthy. For the purpose of this study, we have assumed that this interval is part of the species description.

## Supporting information

**S1 Fig. Heatmap of global specimen collections.** Heatmap showing the distribution of specimen collection events in the four analyzed regions over time. The x-axis represents the months of the year, while the y-axis represents the years of collection. The color intensity indicates the number of specimens collected in each time period.
(PDF)

**S2 Fig. Heatmap of specimen collections in Madagascar.** Heatmap showing the distribution of specimen collection events in Madagascar over time. The x-axis represents the months of the year, while the y-axis represents the years of collection. The color intensity indicates the number of specimens collected in each time period.
(PDF)

**S3 Fig. Heatmap of specimen collections in India.** Heatmap showing the distribution of specimen collection events in India over time. The x-axis represents the months of the year, while the y-axis represents the years of collection. The color intensity indicates the number of specimens collected in each time period.
(PDF)

**S4 Fig. Heatmap of specimen collections in Melanesia.** Heatmap showing the distribution of specimen collection events in Melanesia over time. The x-axis represents the months of the year, while the y-axis represents the years of collection. The color intensity indicates the number of specimens collected in each time period.
(PDF)

**S5 Fig. Heatmap of specimen collections in Ecuador.** Heatmap showing the distribution of specimen collection events in Ecuador over time. The x-axis represents the months of the year, while the y-axis represents the years of collection. The color intensity indicates the number of specimens collected in each time period.
(PDF)

**S6 Fig. Residual diagnostics from the DHARMa R package for the fitted models.**
(PDF)

**S7 Fig. Significant effect plots of variables on description *sensu stricto*.** Colored boxes differentiate the regions. Gray = global, Purple = Ecuador, Orange = Madagascar, Green = India, and Blue = Melanesia.
(PDF)

**S8 Fig. Temporal evolution of variables in Ecuador.** (**a**) Overall description process (black), collection (blue), and description *sensu stricto* (green), (**b**) Nº of authors in each species description (mean), (**c**) Number of published taxonomic papers (count), (**d**) Taxonomic papers that include genetics (percentage), (**e**) Nº of species described each year (count), (**f**) Nº of species described in each paper (mean), (**g**) Nº of type specimens in the type series (mean), (**h**) Nº of field trips conducted to collect the entire type series (mean).
(PDF)

**S9 Fig. Temporal evolution of variables in India.** (**a**) Overall description process (black), collection (blue), and description *sensu stricto* (green), (**b**) Nº of authors in each species description (mean), (**c**) Number of published taxonomic papers (count), (**d**) Taxonomic papers that include genetics (percentage), (**e**) Nº of species described each year (count), (**f**) Nº of species described in each paper (mean), (**g**) Nº of type specimens in the type series (mean), (**h**) Nº of field trips conducted to collect the entire type series (mean).
(PDF)

**S10 Fig. Temporal evolution of variables in Madagascar.** (**a**) Overall description process (black), collection (blue), and description *sensu stricto* (green), (**b**) Nº of authors in each species description (mean), (**c**) Number of published taxonomic papers (count), (**d**) Taxonomic papers that include genetics (percentage), (**e**) Nº of species described each year (count), (**f**) Nº of species described in each paper (mean), (**g**) Nº of type specimens in the type series (mean), (**h**) Nº of field trips conducted to collect the entire type series (mean).
(PDF)

**S11 Fig. Temporal evolution of variables in Melanesia.** (**a**) Overall description process (black), collection (blue), and description *sensu stricto* (green), (**b**) Nº of authors in each species description (mean), (**c**) Number of published taxonomic papers (count), (**d**) Taxonomic papers that include genetics (percentage), (**e**) Nº of species described each year (count), (**f**) Nº of species described in each paper (mean), (**g**) Nº of type specimens in the type series (mean), (**h**) Nº of field trips conducted to collect the entire type series (mean).
(PDF)

**S1 Table. Collection and publication dates for each species and region extracted from the literature.**
(XLSX)

**S2 Table. Lag times of collection (MEX_MIN_YEAR_COLPUB), description *sensu stricto* (MIN_YEAR_COLPUB) and overall process (MAX_YEAR_COLPUB) for each species and region. Categorical and numerical variables used in models are also included.**
(XLSX)

**S3 Table. Statistical comparison of AIC values between empiric (fixed + random variables) and null (random variables only) models.**
(PDF)

**S4 Table. p-values from the GLMM models and evaluation metrics for global and regional analyses.**
(PDF)

## Acknowledgments

We want to acknowledge colleagues who provided fruitful discussions on this issue in the past years, specially Matthijs P. van den Burg and Santiago Castroviejo-Fisher. We are grateful to Ernesto Recuero for his critical reading of the manuscript. We thank the academic editor, Alex Slavenko, as well as Paul Oliver, and one anonymous reviewer for their valuable comments and suggestions.

## Author contributions

**Conceptualization:** Albert Carné, Alberto Sánchez-Vialas, Ignacio De la Riva.

**Data curation:** Albert Carné, Alberto Sánchez-Vialas, Claudia Lansac, Miriam Moreno-Orosa.

**Formal analysis:** Albert Carné.

**Investigation:** Albert Carné, Alberto Sánchez-Vialas, Claudia Lansac, Miriam Moreno-Orosa, Ignacio De la Riva.

**Methodology:** Albert Carné, Alberto Sánchez-Vialas, Ignacio De la Riva.

**Project administration:** Albert Carné.

**Supervision:** Ignacio De la Riva.

**Validation:** Albert Carné, Alberto Sánchez-Vialas, Claudia Lansac, Miriam Moreno-Orosa, Ignacio De la Riva.

**Visualization:** Albert Carné, Claudia Lansac.

**Writing – original draft:** Albert Carné.

**Writing – review & editing:** Albert Carné, Alberto Sánchez-Vialas, Claudia Lansac, Miriam Moreno-Orosa, Ignacio De la Riva.

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
