## [Decision Letter · Decision Letter 0]

27 Jun 2025

PONE-D-25-19328Croaking for haste: How long does it take to describe a frog species since its discovery?PLOS ONE?

Dear Dr. De la Riva,

Thank you for submitting your manuscript to PLOS ONE. After careful consideration, we feel that it has merit but does not fully meet PLOS ONE’s publication criteria as it currently stands. Therefore, we invite you to submit a revised version of the manuscript that addresses the points raised during the review process.

We look forward to receiving your revised manuscript.

Kind regards,

Alex Slavenko

Academic Editor

PLOS ONE

**Journal Requirements:**

1. When submitting your revision, we need you to address these additional requirements. Please ensure that your manuscript meets PLOS ONE's style requirements, including those for file naming. The PLOS ONE style templates can be found at https://journals.plos.org/plosone/s/file?id=wjVg/PLOSOne_formatting_sample_main_body.pdf and https://journals.plos.org/plosone/s/file?id=ba62/PLOSOne_formatting_sample_title_authors_affiliations.pdf 2. Thank you for stating the following financial disclosure: AC was funded by MCIN/AEI/10.13039/501100011033, contract for industrial doctorates aid DIN2021-011964.   Please state what role the funders took in the study.  If the funders had no role, please state: "The funders had no role in study design, data collection and analysis, decision to publish, or preparation of the manuscript." If this statement is not correct you must amend it as needed. Please include this amended Role of Funder statement in your cover letter; we will change the online submission form on your behalf. 3. When completing the data availability statement of the submission form, you indicated that you will make your data available on acceptance. We strongly recommend all authors decide on a data sharing plan before acceptance, as the process can be lengthy and hold up publication timelines. Please note that, though access restrictions are acceptable now, your entire data will need to be made freely accessible if your manuscript is accepted for publication. This policy applies to all data except where public deposition would breach compliance with the protocol approved by your research ethics board. If you are unable to adhere to our open data policy, please kindly revise your statement to explain your reasoning and we will seek the editor's input on an exemption. Please be assured that, once you have provided your new statement, the assessment of your exemption will not hold up the peer review process. 4. We note that Figure 1a in your submission contain map images which may be copyrighted. All PLOS content is published under the Creative Commons Attribution License (CC BY 4.0), which means that the manuscript, images, and Supporting Information files will be freely available online, and any third party is permitted to access, download, copy, distribute, and use these materials in any way, even commercially, with proper attribution. For these reasons, we cannot publish previously copyrighted maps or satellite images created using proprietary data, such as Google software (Google Maps, Street View, and Earth). For more information, see our copyright guidelines: http://journals.plos.org/plosone/s/licenses-and-copyright. We require you to either present written permission from the copyright holder to publish these figures specifically under the CC BY 4.0 license, or remove the figures from your submission: a. You may seek permission from the original copyright holder of Figure 1a to publish the content specifically under the CC BY 4.0 license.   We recommend that you contact the original copyright holder with the Content Permission Form (http://journals.plos.org/plosone/s/file?id=7c09/content-permission-form.pdf) and the following text:“I request permission for the open-access journal PLOS ONE to publish XXX under the Creative Commons Attribution License (CCAL) CC BY 4.0 (http://creativecommons.org/licenses/by/4.0/). Please be aware that this license allows unrestricted use and distribution, even commercially, by third parties. Please reply and provide explicit written permission to publish XXX under a CC BY license and complete the attached form.” Please upload the completed Content Permission Form or other proof of granted permissions as an "Other" file with your submission. In the figure caption of the copyrighted figure, please include the following text: “Reprinted from [ref] under a CC BY license, with permission from [name of publisher], original copyright [original copyright year].” b. If you are unable to obtain permission from the original copyright holder to publish these figures under the CC BY 4.0 license or if the copyright holder’s requirements are incompatible with the CC BY 4.0 license, please either i) remove the figure or ii) supply a replacement figure that complies with the CC BY 4.0 license. Please check copyright information on all replacement figures and update the figure caption with source information. If applicable, please specify in the figure caption text when a figure is similar but not identical to the original image and is therefore for illustrative purposes only.The following resources for replacing copyrighted map figures may be helpful: USGS National Map Viewer (public domain): http://viewer.nationalmap.gov/viewer/The Gateway to Astronaut Photography of Earth (public domain): http://eol.jsc.nasa.gov/sseop/clickmap/Maps at the CIA (public domain): https://www.cia.gov/library/publications/the-world-factbook/index.html and https://www.cia.gov/library/publications/cia-maps-publications/index.htmlNASA Earth Observatory (public domain): http://earthobservatory.nasa.gov/Landsat:
http://landsat.visibleearth.nasa.gov/USGS EROS (Earth Resources Observatory and Science (EROS) Center) (public domain): http://eros.usgs.gov/#Natural Earth (public domain): http://www.naturalearthdata.com/ ?>**Additional Editor Comments:**

I have now received two reviews for this manuscript. I agree with both reviewers that this manuscript presents an interesting analysis of an impressive dataset, and has some solid results and discussions on a very timely research matter. However, I also share some of reviewer 2's concerns. Some of these are mostly editorial (i.e., the manuscript is quite long and overly detailed in some cases and can be streamlined, as per the excellent suggestion of reviewer 2) and some are related to the content. I agree that some statements made are overly broad and sweeping, and the language can be toned down at times. I also personally have some concerns about advocating fast-track taxonomy and wikification as solutions to the issue. I agree with reviewer 2, also based on my personal experience, that in many cases doing taxonomy properly just takes a whole lot of time, and without investing more resources into there's not much that can be done to speed it up.

Overall, I think this is a solid manuscript with a lot of potential, and I encourage the authors to read and adress the reviewer comments carefully if they wish to submit a revised version.

Reviewers' comments:

**Comments to the Author**

1. Is the manuscript technically sound, and do the data support the conclusions?

Reviewer #1: Yes

Reviewer #2: Partly

2. Has the statistical analysis been performed appropriately and rigorously?

Reviewer #1: I Don't Know

Reviewer #2: I Don't Know

3. Have the authors made all data underlying the findings in their manuscript fully available?

Reviewer #1: Yes

Reviewer #2: Yes

4. Is the manuscript presented in an intelligible fashion and written in standard English?

Reviewer #1: Yes

Reviewer #2: Yes

**Reviewer #1:**  General Assessment:

This manuscript presents a well-conceived and original study based on a remarkable dataset. To the best of my knowledge, it is the first work to quantitatively assess the full duration of the taxonomic description process—from the initial collection of specimens to the eventual publication of species descriptions. The research addresses an important and timely issue in biodiversity science and has the potential to make a significant contribution to the field.

The authors’ approach to categorizing and formalizing their dataset is methodologically rigorous, transparent, and well justified. Their assumptions are clearly detailed and appear reasonable and reproducible. However, as I do not have expertise in statistics, I am not in a position to assess the appropriateness or robustness of the statistical tools and models employed in the analysis.

The discussion is pertinent and well articulated. I find the authors’ conclusions and recommendations compelling, especially their suggestions for streamlining the taxonomic workflow to enhance the pace of species inventory. That said, I am not certain that there is a clear consensus within the taxonomic community on some of the proposed recommendations, which highlights the value of this work in stimulating a constructive debate around best practices.

Minor Comments:

I would appreciate it if the authors could briefly explain, in the Materials and Methods section, the meaning and relevance of the statistical indices reported in Table 1 (e.g., Skew, MAD, Kurtosis). For readers who are not trained statisticians, a concise clarification of what each index measures and how it contributes to the analysis would improve accessibility.

Conclusion:

In summary, this is a well-executed, well-written, and highly informative article. It presents an original contribution to the literature on taxonomic research. I have no major criticisms and only a single minor suggestion for clarification. I recommend this manuscript for publication after this small revision is addressed.

**Reviewer #2:**  This paper is an interesting read that provides information on the temporal distribution of key steps on the road to describing frog species in tropical areas. I think with careful revision it could be worth publishing. I have made some more detailed comments on a word version of the MS, but also make some broader comments here.

Tone of writing - Overall I suggest there may be room to be a bit more tempered in some of the claims made. As examples - for vertebrates (and especially terrestrial verts) it is defendable to argue that we do have moderately good framework for understanding patterns in many areas (i.e the existing framework is not a complete disaster). In developing countries, the claim that naming species leads to their protection is contestable at best, as conservation, let alone conservation of small brown frogs, is low on the priority list. In these places in the short-term individual species focused conservation is also not likely to get off the ground – broader region-based approaches are probably more likely to get traction. This is not to say the taxonomy is not important – it totally is and can feed into the above – but it is not the be all and end all to getting conservation happening in tropical countries. Basically - I suggest temper the tone a little in places.

Results – I think these can be rationalised a bit. They are quite long at the moment, but many cases are really just verbal summaries of the tables. Try and cut them down. For instance instead of by region summaries – just say the global trends, and note where regions differ from this. The discussion of months where types where collected is not interesting and should be largely or completely cut.

There is a high risk that any differences are largely idiosyncratic so I would strongly suggest focusing on the global trends.

Discussion

Following from the above the discussion is my opinion too long, and includes a lot of general thoughts that are not well connected with the key results and/or quite sweeping. I would suggest this needs a thorough rewrite to make it much more focused.

I would also strongly suggest not framing this around crisis and solution – we have been hearing this for 20 years now (at least) and it has not achieved a great deal. For what it is worth, here is my summary based on working in Melanesia, which likely has over 800 frog species and about 5 active taxonomists (none of which are paid full time to be taxonomists). First there is not going to be a magic bullet, my uni does see describing frogs as a priority activity for me, and neither do most people in Melanesia. That said I do think it is of great value and agree with the authors. So what we can do however is look at the resources we have, the shape the job to be done, and try and prioritise things on criteria such as which things are likely to be most threatened, which descriptions are easiest, and which descriptions will have most impact. If we do this smart we may then be able to attract support to keep working on the job. When people are really keen to work in this space we can support and enable them, while trying to be honest about their job prospects, but we also can’t engineer a trained and employed New Guinea frog taxonomist out of thin air. I am not against new technologies at all, but with 5 people in the space (three which are over 60) we have to be realistic about what we can get done. I would suggest a more focused discussion about how to prioritise resources and potentially to advocate for a few more could be a useful thing that can be done here.

I have noted in the marked up word doc some other cases where sweeping statements are good in principle, but in practice can pose issues, or need to be much more fully fleshed out.

Areas for consideration with respect to interpretation and analyses

The time from collection to description is actually shorter than I expected. For me this emphasises that we really need to keep collecting (as we and many other have stated in lit). Most new species are relatively newly collected, indicating that the reference libraries of life (museums) have collections that are far from complete. This point has been made, but given increasing hurdles to fieldwork and vouchering, it does not hurt to make it again. If we don't keep collecting - especially in poorly known areas, we are not gunna get the job done.

Is lag time correlated with the number of active taxonomists? I know in Melanesia there are only four living scientists that have described more than five species in this region. I suggest an analysis of how numbers of active authors may influence description speed. This could also illustrate how few people are working on some diverse faunas which would show how tenuous all this knowledge actually is.

I would suggest the 11.3 year lag is likely an underestimate going forward, as the sample is biased towards frogs that are being described. The frogs that remain undescribed are increasingly, the hard ones – complexes of confusing or morphologically similar species that are difficult to resolve. One could speculate that lag time may increase as we increasingly get stuck on these species. But only put this in if space opens up – more important in my opinion to rationalise and focus discussion around some of the points raised earlier.

Keywords

Crisis is misspelt, order words alphabetically

**Do you want your identity to be public for this peer review?** For information about this choice, including consent withdrawal, please see our Privacy Policy

Reviewer #1: No

Reviewer #2: **Yes:** Paul Oliver

---

## [Author Response · Author response to Decision Letter 1]

16 Aug 2025

We uploaded a formatted PDF ("Response_to_reviewers") with detailed responses to the editor and reviewers. However, we paste the non-formatted version here:

MS #PONE-D-25-19328 Croaking for haste: How long does it take to describe a frog species since its discovery?

Authors’ replies to the comments of the Editor and Reviewers

We thank the editor and reviewers for their positive and constructive comments. We have followed almost all of these. We reply to each comment in the following pages and explain how we dealt with them during the manuscript revision. Please, find below our responses (in black) to the editor's and reviewers' comments (in blue). Each response is preceded by "-Response" for easy identification. All changes in the newly uploaded manuscript have been highlighted using track changes. We hope that the manuscript is now suitable for publication in PLOS ONE.

PLOS ONE

Dear Dr. De la Riva,

Thank you for submitting your manuscript to PLOS ONE. After careful consideration, we feel that it has merit but does not fully meet PLOS ONE’s publication criteria as it currently stands. Therefore, we invite you to submit a revised version of the manuscript that addresses the points raised during the review process.

We look forward to receiving your revised manuscript.

Kind regards,

Alex Slavenko

Academic Editor

PLOS ONE

Journal Requirements:

---Response: We have checked and our files meet PLOS ONE’S style requirements.

2. Thank you for stating the following financial disclosure: AC was funded by MCIN/AEI/10.13039/501100011033, contract for industrial doctorates aid DIN2021-011964.

---Response: We have added the suggested information and included it at the end of the cover letter.

--- Before it read: AC was funded by MCIN/AEI/10.13039/501100011033, contract for industrial doctorates aid DIN2021-011964.

--- Now it reads: AC was funded by MCIN/AEI/10.13039/501100011033, contract for industrial doctorates aid DIN2021-011964. The funders had no role in study design, data collection and analysis, decision to publish, or preparation of the manuscript.

---Response: Perhaps we selected this option in error. All data were already included in the initial submission and were uploaded as supplementary files.

4. We note that Figure 1a in your submission contain map images which may be copyrighted. All PLOS content is published under the Creative Commons Attribution License (CC BY 4.0), which means that the manuscript, images, and Supporting Information files will be freely available online, and any third party is permitted to access, download, copy, distribute, and use these materials in any way, even commercially, with proper attribution. For these reasons, we cannot publish previously copyrighted maps or satellite images created using proprietary data, such as Google software (Google Maps, Street View, and Earth). For more information, see our copyright guidelines: http://journals.plos.org/plosone/s/licenses-and-copyright.

We require you to either present written permission from the copyright holder to publish these figures specifically under the CC BY 4.0 license, or remove the figures from your submission:

a. You may seek permission from the original copyright holder of Figure 1a to publish the content specifically under the CC BY 4.0 license.

---Response: The data we used to create the map comes from (https://public.opendatasoft.com/explore/dataset/world-administrative-boundaries/information/) and is licensed under the Open Government Licence v3.0, which is compatible with the Creative Commons Attribution License (CC BY 4.0). We have included this information in the figure caption.

--- Before it read: Time lags and trends in anuran amphibian species descriptions. (a) Cumulative number of frog species described over time. (b) Time lags in the description process. (c) Average description speed per year. Green: description sensu stricto, blue: collection, gray: overall process.

--- Now it reads: Time lags and trends in anuran amphibian species descriptions. (a) Cumulative number of frog species described over time. (b) Time lags in the description process. (c) Average description speed per year. Green: description sensu stricto, blue: collection, gray: overall process. The world map showing country boundaries was obtained from OpenDataSoft (https://public.opendatasoft.com/explore/dataset/world-administrative-boundaries/information/) and is licensed under the Open Government Licence v3.0, which is compatible with the Creative Commons Attribution License (CC BY 4.0).

Additional Editor Comments:

I have now received two reviews for this manuscript. I agree with both reviewers that this manuscript presents an interesting analysis of an impressive dataset, and has some solid results and discussions on a very timely research matter. However, I also share some of reviewer 2's concerns. Some of these are mostly editorial (i.e., the manuscript is quite long and overly detailed in some cases and can be streamlined, as per the excellent suggestion of reviewer 2) and some are related to the content. I agree that some statements made are overly broad and sweeping, and the language can be toned down at times. I also personally have some concerns about advocating fast-track taxonomy and wikification as solutions to the issue. I agree with reviewer 2, also based on my personal experience, that in many cases doing taxonomy properly just takes a whole lot of time, and without investing more resources into there's not much that can be done to speed it up.

---Response: We acknowledge that the manuscript in its original form was quite long and detailed. Following Reviewer #2’s suggestions, we have reduced the length of some sections by focusing on the general patterns, which has shortened the manuscript. Additionally, following the editor’s and Reviewer #2’s suggestions, we have toned down the language throughout the manuscript. Finally, we have placed greater emphasis on the need for increased economic resources for taxonomy.

Overall, I think this is a solid manuscript with a lot of potential, and I encourage the authors to read and adress the reviewer comments carefully if they wish to submit a revised version.

Reviewers' comments:

Reviewer's Responses to Questions

Comments to the Author

1. Is the manuscript technically sound, and do the data support the conclusions?

Reviewer #1: Yes

Reviewer #2: Partly

2. Has the statistical analysis been performed appropriately and rigorously?

Reviewer #1: I Don't Know

Reviewer #2: I Don't Know

3. Have the authors made all data underlying the findings in their manuscript fully available?

Reviewer #1: Yes

Reviewer #2: Yes

4. Is the manuscript presented in an intelligible fashion and written in standard English?

Reviewer #1: Yes

Reviewer #2: Yes

5. Review Comments to the Author

Reviewer #1: General Assessment:

This manuscript presents a well-conceived and original study based on a remarkable dataset. To the best of my knowledge, it is the first work to quantitatively assess the full duration of the taxonomic description process—from the initial collection of specimens to the eventual publication of species descriptions. The research addresses an important and timely issue in biodiversity science and has the potential to make a significant contribution to the field.

The authors’ approach to categorizing and formalizing their dataset is methodologically rigorous, transparent, and well justified. Their assumptions are clearly detailed and appear reasonable and reproducible. However, as I do not have expertise in statistics, I am not in a position to assess the appropriateness or robustness of the statistical tools and models employed in the analysis.

The discussion is pertinent and well articulated. I find the authors’ conclusions and recommendations compelling, especially their suggestions for streamlining the taxonomic workflow to enhance the pace of species inventory. That said, I am not certain that there is a clear consensus within the taxonomic community on some of the proposed recommendations, which highlights the value of this work in stimulating a constructive debate around best practices.

Minor Comments:

I would appreciate it if the authors could briefly explain, in the Materials and Methods section, the meaning and relevance of the statistical indices reported in Table 1 (e.g., Skew, MAD, Kurtosis). For readers who are not trained statisticians, a concise clarification of what each index measures and how it contributes to the analysis would improve accessibility.

---Response: We thank the reviewer for their thoughtful and constructive comments, and for their careful reading of our manuscript.

Some of these metrics are already indirectly described in lines 283–291, where we summarize the values obtained for each (or nearly each) metric. However, we agree with Reviewer #1 that a brief explanation of the metrics would improv

---

## [Decision Letter · Decision Letter 1]

27 Oct 2025

PONE-D-25-19328R1Croaking for haste: How long does it take to describe a frog species since its discovery?PLOS ONE?

Dear Dr. De la Riva,

Thank you for submitting your manuscript to PLOS ONE. After careful consideration, we feel that it has merit but does not fully meet PLOS ONE’s publication criteria as it currently stands. Therefore, we invite you to submit a revised version of the manuscript that addresses the points raised during the review process.

We look forward to receiving your revised manuscript.

Kind regards,

Alex Slavenko

Academic Editor

PLOS ONE

Journal Requirements:

Additional Editor Comments:

The manuscript has now been seen again by reviewer 2, who was the more critical of the original reviewers. As you will see, this reviewer still has some issues with the interpretation of the data and some of the arguments the authors make. While I generally agree that there is perhaps an over-simplification of the issues plaguing taxonomy and leading to long delays between discovery and description of new species, overall I think the manuscript presents strong and valuable data and has many interesting things to say. I would urge the authors to again possibly rephrase some of the discussion talking points - but ultimately, if you decide not to and that you would like to maintain your discussion as is, it will not get in the way of publication.

Reviewer's Responses to Questions

**Comments to the Author**

Reviewer #2: All comments have been addressed

2. Is the manuscript technically sound, and do the data support the conclusions?

Reviewer #2: Yes

3. Has the statistical analysis been performed appropriately and rigorously?

Reviewer #2: Yes

4. Have the authors made all data underlying the findings in their manuscript fully available?

Reviewer #2: Yes

5. Is the manuscript presented in an intelligible fashion and written in standard English?

Reviewer #2: Yes

Reviewer #2: I have now reviewed this manuscript for a second time. I think the changes that have been made greatly improve clarity and readability, and I appreciate that the authors have taken time to respond to all suggestions.

I have gone through and tried to make some comments on the revised pdf on areas that could be potentially further tidied up.

In general - my opinion is still that discussion and small parts of the introduction are not very convincing and in places are not really providing helpful solutions that are going to address the issues in place. I would be the cutting down the discussion of how we can use novel approaches to speed up taxonomy, and focusing in the key messages that your data supports - a) it still takes a while to get things described and this lag might be increasing, b) there is still much we don't know as evidenced by most new species being recently collected, c) hence we need to keep collecting to have any hope of filling these gaps, and d) under-resourced museums and taxonomists are critical to meeting this challenge.

More specifically I'm sorry I did not pick this up earlier but I think part of the issue here is that dichotomy between fast-track and traditional taxonomy is not very clear or at all convincing (at least to me). Yes we should keep looking at ways to make things more efficient - I agree. But what exactly is being proposed to be cut from a "typical" descriptions to make things quicker? In my experience the deeper you dig into this the harder it becomes to work out what you are going to cut - especially for the god-awful species complexes we increasingly have to work with in groups like frogs - this is where you need more detail to actually understand what you are looking at. yes you can cut some of the detailed descriptive stuff - but in practice this actually often does not take very long and is often very brief anyway - so cutting it will not in fact save enormous amounts of time. There is also a trade off here in making descriptions useful versus quick. This is not an argument to better define fast-track taxonomy, it is argument to cut down reference to it completely as I actually don't think it is in practice really much different that what is being done or a 'solution".

Following on this I will say that my general point in the first review with the summary of the Melanesian frog fauna was that with ~5 people really active in taxonomy, and most of them spending 50-75% of their time doing other things - fast-track taxonomy, wiki-style sharing of data etc etc is not really going to achieve help that much (and indeed may just give the small number of people working the problem a world more work to do). The issue is capacity. I do understand that part of the idea here is that by emphasising the new approaches it has the potential attract new support and resources to this work, however I think in practice most of what is suggested is no game changer, and the only real game changer here is more funding and more taxonomists (or even just funding for the taxonomists that exist to do taxonomy!). I concede the story might be different in other areas that have more workers with potentially better resourcing that I don't know so well.

To give a final bit of clarity and perhaps a warning about tone I also quote one of the response to review comments here:

"Our point is that if 64% of species were described more than five years after collection, this suggests that many new species are either not recognized in the field or are not prioritized for description. In our view, if a species is clearly recognized as new in the field, it should not take ten years or more for its formal description. We acknowledge that delays can occur for many reasons, but taking a decade (or much more) to describe something identified as new from the outset seems excessive, and the less parsimonious explanation."

To be honest, this comes across as a staggeringly naive thing to say - I know people who have been working on taxonomy of groups in both Australia and New Guinea that have dozens to hundreds of taxa that they are desperate to describe, but due to the massive amount work involved in even basic characterisation and diagnosis of these species, the sheer numbers of new species that need to be described, and the fact that more often than not no-one is paying them to do this work, and the fact they want to sleep at night - the descriptive process is taking decades or more.

I guess my general point here is that yes we should absolutely be looking for ways to characterise issues and push things forwards and speed things up when we can, but these impediments and issues exist for many and complicated reasons and people have been thinking about it for decades. So I think focus more on the data you have and what it shows, and acknowledge that the main impediments are not procedural, but time and resources (with perhaps a large dash of legislation and bureaucracy for extra flavour).

I will leave it to editor and authors as to how much they want to act on this - I think the data as presented is fair and reasonable and summaries are interesting. And hopefully this is review helpful it is genuinely meant as to be - it is an interesting paper, and certainly question of how well we are doing in documenting the fauna of these poorly-known regions and groups is an important one to keep track of.

Kind regards

Paul

**Do you want your identity to be public for this peer review?** For information about this choice, including consent withdrawal, please see our Privacy Policy

Reviewer #2: **Yes:** Paul Oliver

---

## [Author Response · Author response to Decision Letter 2]

15 Dec 2025

We have attached the responses to the editor and reviewers in a PDF document titled “0_Response_to_reviewers.pdf.”

---

## [Editor Report · Decision Letter 2]

5 Jan 2026

Croaking for haste: How long does it take to describe a frog species since its discovery?

PONE-D-25-19328R2

Dear Dr. De la Riva,

We’re pleased to inform you that your manuscript has been judged scientifically suitable for publication and will be formally accepted for publication once it meets all outstanding technical requirements.

Kind regards,

Alex Slavenko

Academic Editor

PLOS One
---

## [Editor Report · Acceptance letter]

PONE-D-25-19328R2

PLOS One

Dear Dr. De la Riva,

I'm pleased to inform you that your manuscript has been deemed suitable for publication in PLOS One. Congratulations! Your manuscript is now being handed over to our production team.

Kind regards,

on behalf of

Dr. Alex Slavenko

Academic Editor

PLOS One